# Determination of radiocarbon in environmental objects

**Almira Raimkanova**, **Ainur Mamyrbayeva**, **Almira Aidarkhanova***,
**Assan Aidarkhanov, Aidana Sarsenova, Zhanna Tleukanova**

Institute of Radiation Safety and Ecology, National Nuclear Center of the Republic of Kazakhstan, Kurchatov, Kazakhstan

* almira@nnc.kz

## Abstract

This article presents the results of radiocarbon distribution in natural components of 'Degelen' test location. It was used to conduct underground nuclear tests at the Semipalatinsk Test Site. Near-entry areas with water inflows were selected for research. Soil, plant and water samples were taken, in which the radiocarbon content was determined. The $^{14}C$ activity concentration was determined using a highly sensitive alpha-beta radiometer SL-300 (ISO 13162, 2021). Sample preparation was carried out using a modern automated system Pyrolyser-6 Trio (Raddec International Ltd, UK). In the course of research undertaken, radiocarbon was revealed to be nonuniformly distributed in environmental objects. The effectiveness of Pyrolyser-6 Trio calcination and ashing system for $^{14}C$ determination in environmental matrices has been demonstrated.

## 1. Introduction

Carbon on Earth is found in the atmosphere, flora, fauna, non-living organic matter, fossil fuel, rock and is dissolved in the ocean. Carbon is the sixth most abundant element in the universe after hydrogen, helium, oxygen, neon and nitrogen. A concern about increasing levels of carbon dioxide in the atmosphere has led to a rise in public and scientific interest in the global carbon cycle [1]. The soils of the planet contain twice as much carbon as in the atmosphere producing most of the flux of carbon emissions into the atmosphere by microbial decomposition of soil organic carbon, which currently balances the net flux of carbon entering terrestrial ecosystems as a result of photosynthesis in plants. A change in the dynamics of soil organic carbon can have a significant impact on the atmospheric concentration of $CO_2$, the global carbon cycle and, consequently, on the Earth's climate system [2,3]. Tracing radiocarbon (natural or due to nuclear tests) in terrestrial ecosystems can be a powerful tool for studying the dynamics of soil organic carbon [4]. Natural radiocarbon is continuously produced in the lower atmosphere when cosmic neutrons interact with nitrogen

**Editor:** Mohamed Y.M. Hanfi, Ural Federal University named after the first President of Russia B N Yeltsin: Ural'skij federal'nyj universitet imeni pervogo Prezidenta Rossii B N El'cina, RUSSIAN FEDERATION

**Data availability statement:** All relevant data are within the manuscript and its Supporting Information files.

**Funding:** This research was funded by the Ministry of Science and Higher Education of the Republic of Kazakhstan (Grant No. BR21881915).

nuclei. In XX century, nuclear weapon tests conducted between 1945 and 1980 became a key anthropogenic source of $^{14}C$. During nuclear tests, $^{14}C$ occurs due to excess neutrons captured by atmospheric nitrogen. The total activity of $^{14}C$ released into the atmosphere over the mentioned period was about $3.5 \cdot 10^8$ GBq [5]. The label of the global content of 'bomb' $^{14}C$ has proved to be a useful indicator for studying the dynamics of soil organic carbon on time scales from several years to decades, since it allows us to estimate how much organic carbon, fixed by photosynthesis, has mixed with the organic component of the soil [6–8]. The paper [9] presents the assessment of $^{14}C$ content in surface soils collected after ground and underground tests at the Nevada National Security Test Site. Concentrations of $^{14}C$ ($319 \pm 9$ pMC) were recorded in soils of underground testing areas, which are approximately 720 Bq/kg. An extremely high content of $^{14}C$ (~1,000–10,000 pMC) was recorded in soils of ground test locations. The authors of these studies assume that $^{14}C$ in the topsoil after ground tests is primarily formed due to neutron activation of the natural soil material in situ, whereas $^{14}C$ in the topsoil from locations of underground tests may originate from the recondensed material of solid particles, or from soil activation.

From this point of view, the study of $^{14}C$ distribution in the components of natural environment is of interest in the Semipalatinsk Test Site area (STS). Most of radioactive contamination at STS was caused by ground and underground tests. One of the sites where nuclear tests were conducted is 'Degelen' test location. The site is located in the same-name mountain range, which represents a dome-shaped elevation of 17–18 km in diameter. The total area is ~300 km². The underground nuclear tests were conducted in tunnels. A tunnel was a horizontal mine working of several hundred meters to 2 kilometers long. Altogether, 1961–1989, 209 nuclear tests in 181 tunnels were conducted. Near–entry sites with water streams are the most contaminated at 'Degelen' test site. The main mechanism of a radionuclide transfer outside the mountain range is the migration through waterways that are in the impact zone of tunnel water streams. By now, quite a few studies have been carried out [10–13] dedicated to radioactive contamination of the natural ecosystems of the 'Degelen' test location with such artificial radionuclides as $^3H$, $^{137}Cs$, $^{90}Sr$, $^{241}Am$, $^{239+240}Pu$. However, so far, radiocarbon levels for the purpose of radiation control and radioecological monitoring have not been studied in the area of the 'Degelen' site.

Important analytical approaches to determine the activity of $^{14}C$ in environmental matrices are: (1) sample oxidation to convert carbon species ($^{12}C$, $^{13}C$ and $^{14}C$) into $CO_2$ followed by graphitization and determination of $^{14}C$ content in an accelerator mass spectrometer (AMS) [14–16], (2) sample oxidation to convert carbon species to a $CO_2$ form, then the process of benzene synthesis followed by counting with a liquid scintillation spectrometer (LSS) [17,18] and (3) sample oxidation and counting with a LSS [19–21]. The first two techniques yield highly precise results making it possible to measure very low levels of $^{14}C$ activity in environmental samples and, consequently, are often used for $^{14}C$ dating. However, the sample preparation procedure is complex and lasting, and the analysis is costly, which hampers the analysis of a great many samples using these techniques. The environmental monitoring program for radioactively contaminated areas requires a great many samples to be analyzed

in a short time, therefore sample oxidation and LSS counting are often used, since the third technique yields results in a relatively short time compared to the other two described above. In this technique, the samples are thermally oxidized at high temperature to convert carbon into $CO_2$, which is then captured by a special sorbent and counted in LSS. As of today, the worldwide average activity concentration of $^{14}C$ for a clean region (an area with the lowest local contamination by anthropogenic sources) is ~242 Bq/kg of C [22]. As reported by the Institute de Radioprotection et de SÛreté Nucléaire [23], the increase in the environmental activity concentration of this radionuclide by an operating nuclear power plant with an annual atmospheric emission from 0.2 to $1 \times 10^{12}$ Bq/year is about 3 Bq/kg. To accurately determine the activity concentration of this radionuclide, it is important to reduce any possible measurement errors or uncertainties. Thus, the main tasks of these studies were to evaluate the effectiveness of the developed methodological approaches for determining the concentration levels of radiocarbon in environmental objects located in radioactively contaminated areas the case of the 'Degelen' test location.

## 2. Materials and techniques

### 2.1. Objects of research

Tritium was chosen as a tracer when selecting research sites because, like radiocarbon, it is produced by neutron activation processes during nuclear tests. Based on the analysis of available data [10] on the maxima of tritium activity concentration, tunnels No. 104, 165, 176, 177, 504 water streams with water seepage (Fig 1) were chosen as research sites at the 'Degelen' test site (Fig 1 in S1 Appendix). Duplicate samples of water, soil and plants were collected in the vicinity of interest water streams in the summer (Fig 2 in S1 Appendix). Water samples were collected to a glass container. The container was filled as much as possible to minimize the exchange of $^{14}C$ with atmospheric $CO_2$, since water samples should not be oxidized to avoid carbon disequilibrium. For duplicate soil and plant sampling, a 1 m*1m site was prepared 2 m away from the water stream. Soil was sampled from the surface layer to 5 cm deep using the 'envelope' technique. The aboveground motley grass parts were cut off from the entire site.

### 2.2. Research methodology

To determine radiocarbon by liquid scintillation counting, samples were prepared with an automated system Pyrolyser-6 Trio (Raddec International Ltd, UK) (Fig 3 in S1 Appendix). The system consists of six quart-lined tubes that pass through three adjacent ovens: sample zone, middle zone and catalyst zone (Fig 2). Quartz glass work tubes of 30 mm inner diameter are loaded into these tubes, tapering at one end to accommodate a catalyst made of 0.5% aluminum oxide and platinum. The samples are loaded into quartz glass boats and fed into each of the work tubes through an air intake socket pipe or an air/oxygen mixture. The end jack made of borosilicate is attached to a bubbler filled with an absorber. 0.1M $HNO_3$ is used as an absorber of $^3H$, and Carbo-Sorb E (PerkinElmer, USA) consisting of 3-methoxypropylamine is used for $^{14}C$.

The air passes over the sample, which is gradually heated to the maximum temperature using a predetermined cycle of linear temperature change. For complete oxidation, oxygen is fed by the end of the run and mixed with the air. Combustion products of the sample are passed through the catalyst heated to 800°C in the oven of a catalyst zone. Under temperature, radionuclide components are converted into tritiated water (HTO) and radioactive carbon dioxide ($^{14}CO_2$), which are captured by individual $^3H$ and $^{14}C$ bubblers (Fig 4 in S1 Appendix).

The effective combustion of the samples depends on the profile of linear variation applied to the oven of the sample zone. Rapid heating of the sample leads to the lack of combustion control, ineffective capture of radionuclides in bubblers and possible damage to the silicon glass work tubes. The most acceptable heating cycle will depend on the sample matrix and the species of radionuclides in the sample. Pyrolyser-6 Trio has 8 preset programs developed for combusting various types of samples. The first four programs: Organic, Normal, Rapid and Fish are used for combusting environmental

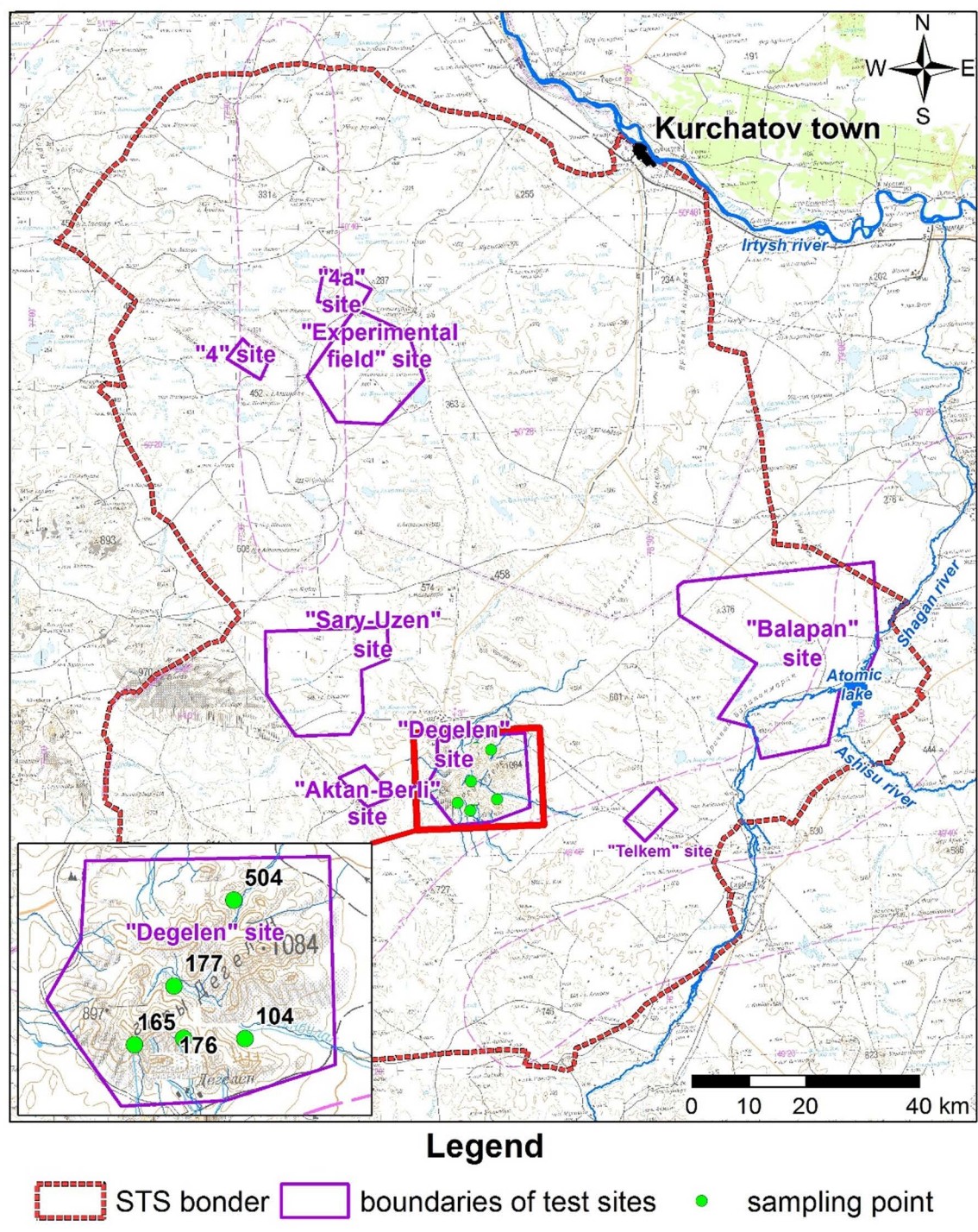

**Fig 1. Sampling points of environment objects.**

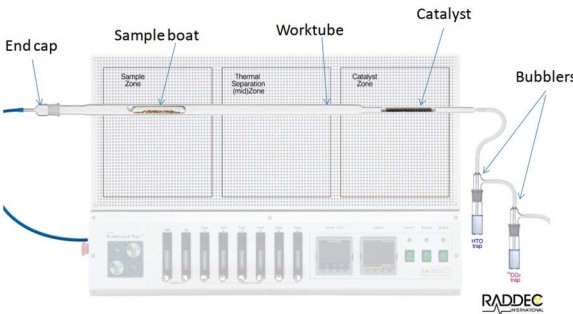

**Fig 2. Pyrolyser-6 Trio schematics.**

matrices, whereas other preset programs: Wet asbestos, Dry asbestos, Graphite and Bioconc are designed to analyze samples from nuclear installation [24]. Soil and water samples were analyzed by means of the Normal program with rising temperature up to 900°C. Plant samples were analyzed with the Organic program. The heating cycles were programmed to ensure controlled and complete oxidation of samples containing high concentrations of organic substances at low temperatures and prevent sudden pressure surges in the work tube [25].

The performance of the combustion procedure (including catalyst performance, monitoring of the total accumulation of background activity levels in Pyrolyser-6 Trio) was evaluated by analyzing certified standards. To assess the yield of $^{14}$C during combustion, an 18.4 kBq/g $^{14}$C sample (V.G. Khlopin Radium Institute, Russia) in the form of an inorganic compound of sodium carbonate was used as a standard. Using a reference standard, model solutions were prepared to preliminarily determine the radiocarbon activity, which was 3,430 ± 515 Bq/l. To evaluate the selective extraction of $^{14}$C and $^{3}$H, standard samples of these radionuclides (SPEC-CHEC, PerkinElmer, USA) in the form of organic compounds were used. The initial activities in SPEC-CHEC samples were: for $^{14}$C = 49.860 dpm/ml ± 2.48%; for $^{3}$H = 48.380 dpm/ml ± 3.67%. As of the time of experiments, the estimated values of $^{14}$C and $^{3}$H activity concentrations were 831,000 Bq/l and 395,300 Bq/l, respectively.

In some cases, the combustion of liquid samples and soil samples rich in organic substances can cause splashing, rapid, uncontrolled formation of gaseous decay products leading to the rupture of quartz work tubes. Accordingly, there are limits on the mass of an assay and its mount. In order to ensure steady combustion of liquid samples, two types of mounts were used: in the form of sand washed with a 10% hydrochloric acid solution and filter paper. 0.5 ml reference solutions were applied to the selected mounts. At the second stage, standard samples (SPEC-CHEC, PerkinElmer, USA) with high activities of $^{14}$C and $^{3}$H were analyzed by combusting acid-washed sand uniformly soaked in these reference solutions in a volume of 0.5 ml. The effective extraction of $^{14}$C during calcination and ashing of Pyrolyser-6 Trio samples may be affected by the absorbing properties of the Carbo-Sorb E reagent. In the course of experiments, the effect of the amount of absorbent fed was studied. The volumes of Carbo-Sorb E injected into the bubblers were 10 ml, 15 ml, 20 ml.

Determination of $^{14}$C activity concentration in obtained samples was carried out using highly sensitive alpha-beta radiometer «SL-300» (Fig 5 in S1 Appendix) [26]. At the initial stage of research, the radiometer was calibrated by efficiency using PerkinElmer calibration standards of $^{14}$C with a quenching curve plotted showing a good factory agreement (Fig 3).

To prepare counting samples, an 8 ml aliquot was taken from $^{14}$C bubblers and mixed with a 12 ml Permaflour E+ scintillation liquid (PerkinElmer, USA). The measurement time of one sample was 300 min. The minimum detectable activity was 6 Bq/kg (Fig 6 in S1 Appendix).

**Fig 3. The curve of $^{14}C$ (Eff) detection efficiency relationship with quenching parameters (TDCR).**

**Table 1. The effect of sample matrix and temperature conditions on chemical yields of $^{14}C$.**

| # | Matrix type | Activity concentration of $^{14}C$, Bq/l Normal program, T=600° C $^{a}AC \pm ^{b}SD$ | Chemical yield of $^{14}C$,% | Activity concentration of $^{14}C$, Bq/l Graphite program, T=900°C AC±SD | Chemical yield of $^{14}C$,% |
|---|---|---|---|---|---|
| 1 | sand | 3 050±460 | 89 | 3 490±520 | 100 |
| 2 | sand | 3 100±470 | 90 | 3 170±480 | 93 |
| 3 | sand | 3 120±470 | 91 | 3 140±470 | 91 |
| 4 | filter | 3 300±500 | 96 | 3 210±480 | 94 |
| 5 | filter | 3 370±510 | 98 | 3 170±480 | 92 |
| 6 | filter | 3 150±470 | 92 | 3 300±500 | 96 |
| Mean: | | – | 93 | – | 94 |

$^{a}AC$ : activity concentrations of 14C

$^{b}SD$ : standard deviation up to 15%

**Table 2. Results of activity concentration measurement of SPEC-CHEC standard samples.**

| Standard samples | Activity concentration, Bq/l | | |
|---|---|---|---|
| | Estimated value AC±SD | $^{3}H$, Pyrolyser-6 Trio AC±SD | $^{14}C$, Pyrolyser-6 Trio AC±SD |
| Standard sample - $^{14}C$ | 831 000±124 000 | ‹ 6 | 748 000±112 000 |
| Standard sample - $^{3}H$ | 395 300±59 000 | 356 000±53 400 | ‹ 6 |

To assess the effect of the sample matrix, temperature and amount of absorbent, the experiments were carried out in triplicate. The following are average values in the tables (Tables 1 and 2). Determination of $^{14}C$ in environmental samples were submitted for quality control. One "blank" sample was added to each batch of analyzed samples (5 samples in each batch). It was prepared in advance from samples collected from territories of "background" content of $^{14}C$. The "blank" sample was analyzed simultaneously with all other samples. It was intended for quality control and to control for possible cross-contamination of samples.

## 3. Results and discussion

The results of experimental studies carried out to determine the effect of sample matrix and temperature condition on $^{14}$C extraction efficiency are listed in Table 1.

It follows from research findings that while analyzing the standard sample of $^{14}$C, sand and filter mounts can be used. In both cases the effective extraction of $^{14}$C occurs. These substances have absorbent properties, which ensures steady combustion of the standard sample. The use of these oxidation process programs shows a high chemical yield of $^{14}$C, which averaged 93–94%.

Table 2 lists the values of the $^{3}$H, $^{14}$C activity concentration in the SPEC-CHEC standard samples.

According to the results of the Table 2, when evaluating the performance of the combustion procedure for standard samples of $^{14}$C and $^{3}$H, values close to the estimated ones were obtained. This indicates a high chemical yield of approximately 90%. In the course of conducted experiments, each absorber was found to be able to absorb only individual components. The solution of 0.1M HNO3 selectively absorbs tritium water (HTO), and Carbo-Sorb E + − radiocarbon dioxide ($^{14}CO_2$).

As a result of the exothermal reaction by the interaction between carbon dioxide and the absorbent Carbo-Sorb E, carbamate. It represents a counting sample when mixed with the scintillator Permaflour E +. It was found that the chemical yield of $^{14}$C rises as the volume of the absorbent increases. Thus, the optimal volume to contain carbon dioxide produced during the catalytic combustion of the sample is 20 ml at most.

Determinations on $^{14}$C in natural components are presented in the Table 3 showing a numerical content of radiocarbon for all test samples other than 1 soil sample.

In the course of conducted studies, the radiocarbon distribution pattern was revealed to be nonuniform in ecosystems of tunnel water streams at 'Degelen' test site. The most contaminated are water streams from tunnels No. 104, 165. The least contaminated is tunnel No. 176. For example, the activity concentration of $^{14}$C in the water ranges from 6,700 ± 1,000–260 ± 40 Bq/l, in plants it varies from 130 ± 20–260 ± 40 Bq/kg, and in soil ᐸ 6 Bq/kg to 90 Bq/kg. The activity concentration values of $^{14}$C in the soil and plants are at the background concentration level [22]. In the previous studies [27], the results with a radiocarbon content of 2,500 ± 380 Bq/kg in the soil samples were obtained. The samples were collected from the epicenter of 'Experimental Field' test site, where ground nuclear tests had been conducted. However, it is worth noting the difference in soil types and microclimates of these sites. The above outputs also prove findings [9] showing that the differences in the content of $^{14}$C in soils from ground and underground test locations are greater than 7 times. The activity concentrations of $^{14}$C in the water of all objects exceed the intervention levels (240 Bq/l) established for drinking water and in most cases exceed a tenfold intervention level [28]. It should be noted that the results obtained are much higher compared to the available reported data on $^{14}$C [29] obtained for the marine environment of the Great Britain, where anthropogenic radiocarbon sources are emissions from nuclear fuel cycle facilities.

Based on the results of the analysis, total radiocarbon for each object was estimated and its distribution in environmental components is presented in % (Fig 4).

Table 3. Content of $^{14}$C in water, plants, soil.

| Sampling place | Activity concentration | | |
| --- | --- | --- | --- |
| | Water, Bq/l | Plants, Bq/kg | Soil, Bq/kg |
| | AC ± SD | AC ± SD | AC ± SD |
| Tunnel No.177 | 3300 ± 500 | 190 ± 30 | 90 ± 13 |
| Tunnel No.104 | 6700 ± 1000 | 130 ± 20 | 50 ± 7 |
| Tunnel No.165 | 6500 ± 980 | 260 ± 40 | 80 ± 12 |
| Tunnel No.504 | 2800 ± 420 | 180 ± 25 | 20 ± 3 |
| Tunnel No.176 | 260 ± 40 | 140 ± 20 | ᐸ 6 |

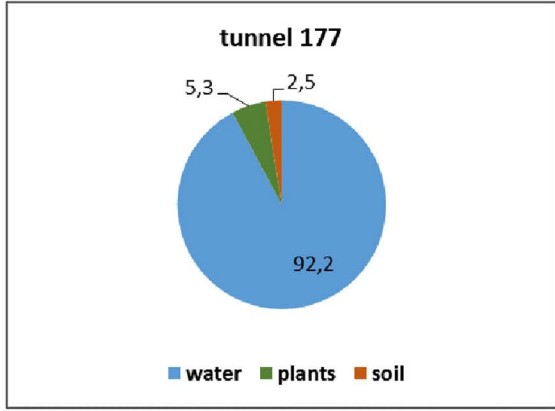

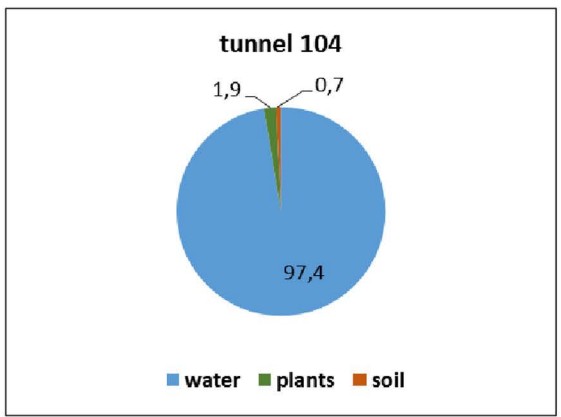

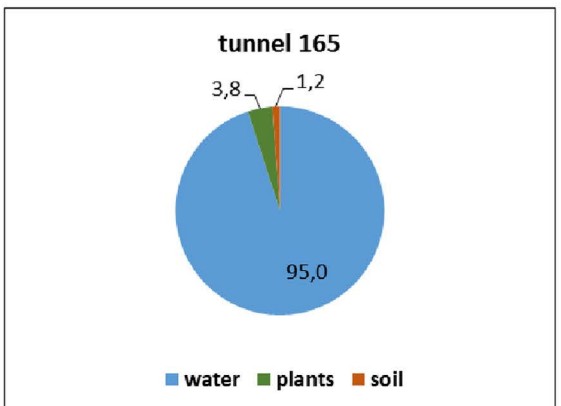

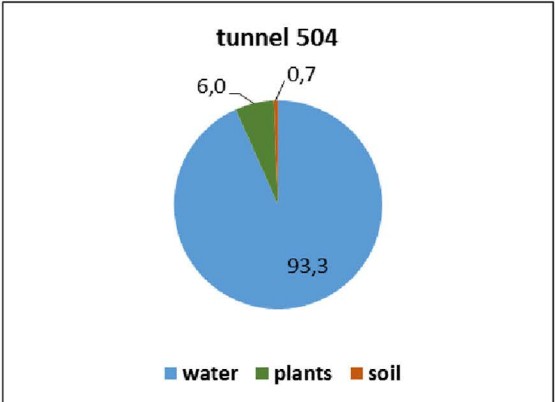

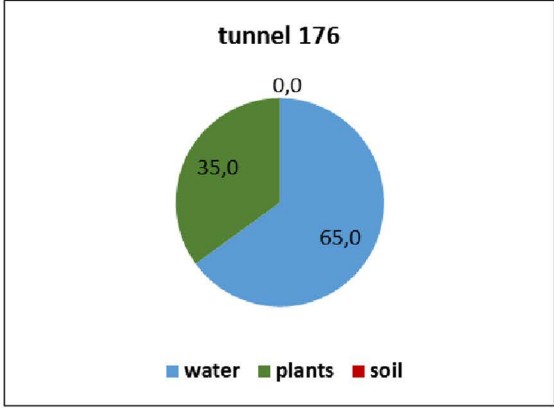

**Fig 4. Percentage distribution of $^{14}$C in natural components.**

In percentage, the predominant content of $^{14}$C for all objects of interest is noted in the water ranging from 92.2 to 97.4%. In plants and soil, in most cases, it ranges from 1.9 to 6.0% and from 0.7 to 2.5%, respectively. A small range in the percentage distribution of $^{14}$C in soils (and in 2 cases – the same values) shows that these ecosystems have a similar nature of $^{14}$C contamination and the same soils type in this area.

Thus, the characteristic distribution for ecosystems of watercourses with a similar $^{14}$C14 contamination mechanism is as follows: water > plants > soil.

## 4. Conclusion

Determination of $^{14}$C concentrations using the Pyrolyser-6 Trio calcination and ashing system required research to ensure reliable data. One of the priority tasks was to use certified standards to control the quality of research results. An important point was to identify the main contributors to sample preparation processes. The accuracy of the radiocarbon analysis was assessed when changing 3 main constituents to affect the outcome most – this is the sample matrix, the temperature conditions and the amount of an absorbent fed. Accounting and control of these parameters guarantees the obtainment of premium and reproducible samples for the liquid scintillation analysis technique. According to experimental findings, some peculiarities of $^{14}$C distribution in the ecosystems of the 'Degelen' test site have been established. The major source of radionuclide intake by other natural components are tunnel water streams, in which maximum concentrations of $^{14}$C are recorded. The research results, conducted using the modern automated Pyrolyser-6 Trio system (Raddec International Ltd, UK), make it possible to optimize the radioecological monitoring program and predict the radioecological situation at the territory of the 'Degelen' test site and beyond.

## Supporting information

**S1 Fig. Appendix.**
(DOCX)

**S1 File. Written permission from the copyright holder.**
(PDF)

**S2 File. Choose a License 260225.**
(PDF)

## Author contributions

**Formal analysis:** Aidana Sarsenova, Zhanna Tleukanova.

**Methodology:** Ainur Mamyrbayeva.

**Supervision:** Assan Aidarkhanov.

**Visualization:** Almira Aidarkhanova.

**Writing – original draft preparation:** Almira Raimkanova.

**Writing – review & editing:** Almira Raimkanova.

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
