## [Decision Letter · Decision Letter 0]

14 Jan 2025

PONE-D-24-39990Determination of radiocarbon in environmental objectsPLOS ONE

Dear Dr. Raimkanova,

Thank you for submitting your manuscript to PLOS ONE. After careful consideration, we feel that it has merit but does not fully meet PLOS ONE’s publication criteria as it currently stands. Therefore, we invite you to submit a revised version of the manuscript that addresses the points raised during the review process.

We look forward to receiving your revised manuscript.

Kind regards,

Mohamed Y.M. Hanfi

Academic Editor

PLOS ONE

Reviewers' comments:

Reviewer's Responses to Questions

**Comments to the Author**

1. Is the manuscript technically sound, and do the data support the conclusions?

Reviewer #1: Yes

Reviewer #2: Yes

2. Has the statistical analysis been performed appropriately and rigorously? 

Reviewer #1: Yes

Reviewer #2: No

3. Have the authors made all data underlying the findings in their manuscript fully available?

Reviewer #1: Yes

Reviewer #2: No

4. Is the manuscript presented in an intelligible fashion and written in standard English?

Reviewer #1: Yes

Reviewer #2: No

5. Review Comments to the Author

Reviewer #1: Comments to the Author

1. Sentence to be rephrased (Page.2, Line no.39)

2. Kindly provide sample collection season

3. What is this HTO (Page.7, Line no.175)

4. In Table 3 Activity Concentration measured Bq/kg for Plants and Soil, but Water Activity Concentration measured Bq/1 . what it indicate

5. In fig.4, What is the significant role shows the same value of 0.7 percentage distribution of 14C in tunnel 104 and tunnel 504

6. How Much distance between the tunnel 104 and tunnel 504

7. Which of the sampling place is more suitable for ecosystems

Reviewer #2: The manuscript provides significant insights into radiocarbon distribution in environmental matrices at the Semipalatinsk Test Site, using the Pyrolyser-6 Trio system and liquid scintillation counting. The methodology is robust, and the results are well-presented. However, minor revisions are necessary. Specifically:

1.Clarify ambiguous terms and improve phrasing for better readability (e.g., rephrasing long sentences in the Results section for clarity).

2.Include additional statistical analysis, such as significance testing, to validate differences in radiocarbon concentrations across sites and matrices.

3.Expand the discussion in the abstract and conclusion to highlight the novelty of using the Pyrolyser-6 Trio system for C-14 determination and its broader implications for environmental monitoring in radioactively contaminated areas.

4.Provide access to raw data or deposit it in a public repository to comply with PLOS ONE’s data availability policies.

These changes will enhance the clarity, statistical rigor, and overall impact of the manuscript without altering its core contributions.

6. PLOS authors have the option to publish the peer review history of their article (what does this mean? ). If published, this will include your full peer review and any attached files.

**Do you want your identity to be public for this peer review?** For information about this choice, including consent withdrawal, please see our Privacy Policy .

Reviewer #1: No

Reviewer #2: No

---

## [Author Response · Author response to Decision Letter 0]

19 Mar 2025

List of corrections

# Comments Explanations New version

Reviewer 1

1. Sentence to be rephrased (Page.2, Line no.39) “Carbon is a foundamental brick of all life forms on Earth. It is present in the atmosphere, flora, fauna, non-living organic matter, fossil fuel, rock and is dissolved in the ocean.”

Corrected

“Carbon on Earth is found in the atmosphere, flora, fauna, non-living organic matter, fossil fuel, rock and is dissolved in the ocean.”

2. Kindly provide sample collection season

Added

“Duplicate samples of water, soil and plants were collected in the vicinity of interest water streams in the summer”

3. What is this HTO (Page.7, Line no.175)

Added

tritiated water (HTO)

4. In Table 3 Activity Concentration measured Bq/kg for Plants and Soil, but Water Activity Concentration measured Bq/1 . what it indicate

Answer to comment

For solid samples, the Activity Concentration was calculated per 1 mass (per 1 kg of plants, 1 kg of soil). For water, the Activity Concentration was calculated per 1 volume of sample (per 1 liter of water).

5. In fig.4, What is the significant role shows the same value of 0.7 percentage distribution of 14C in tunnel 104 and tunnel 504

Added

The same value of 0.7 percentage distribution of 14C in tunnel 104 and tunnel 504 shows only the similar nature of 14C contamination and the same type of soils in the study area.

6. How Much distance between the tunnel 104 and tunnel 504

Answer to comment

The distance between the tunnels is about 9 km, and they are located on opposite sides of the Degelen mountain: the tunnel 104 is located on the south-eastern side, the tunnel 504 - on the northern side.

7. Which of the sampling place is more suitable for ecosystems

Added

L. 328-333

Reviewer 2

8. Clarify ambiguous terms and improve phrasing for better readability (e.g., rephrasing long sentences in the Results section for clarity).

Corrected

9. Include additional statistical analysis, such as significance testing, to validate differences in radiocarbon concentrations across sites and matrices.

Added

L. 246-256

10. Expand the discussion in the abstract and conclusion to highlight the novelty of using the Pyrolyser-6 Trio system for C-14 determination and its broader implications for environmental monitoring in radioactively contaminated areas.

Added

L. 32-40, L. 350-355

11. Provide access to raw data or deposit it in a public repository to comply with PLOS ONE’s data availability policies.

Added

Additional file

No permit for field work was required, since the “National Nuclear Center of the Republic of Kazakhstan” is the authorized body at the territory of the Semipalatinsk nuclear test site, where the “Degelen” test site is located.

---

## [Decision Letter · Decision Letter 1]

2 May 2025

Title - Determination of radiocarbon in environmental objects

PONE-D-24-39990R1

Dear Dr. Almira Aldarkhanova,

We’re pleased to inform you that your manuscript has been judged scientifically suitable for publication and will be formally accepted for publication once it meets all outstanding technical requirements.

Kind regards,

Mohamed Y.M. Hanfi

Academic Editor

PLOS ONE

Additional Editor Comments (optional):

Reviewers' comments:

Reviewer's Responses to Questions

**Comments to the Author**

1. If the authors have adequately addressed your comments raised in a previous round of review and you feel that this manuscript is now acceptable for publication, you may indicate that here to bypass the “Comments to the Author” section, enter your conflict of interest statement in the “Confidential to Editor” section, and submit your "Accept" recommendation.

Reviewer #1: All comments have been addressed

Reviewer #3: (No Response)

2. Is the manuscript technically sound, and do the data support the conclusions?

Reviewer #1: Yes

Reviewer #3: Yes

3. Has the statistical analysis been performed appropriately and rigorously? 

Reviewer #1: Yes

Reviewer #3: Yes

4. Have the authors made all data underlying the findings in their manuscript fully available?

Reviewer #1: Yes

Reviewer #3: Yes

5. Is the manuscript presented in an intelligible fashion and written in standard English?

Reviewer #1: Yes

Reviewer #3: Yes

6. Review Comments to the Author

Reviewer #1: The radiocarbon distribution pattern was non-uniform in ecosystems and this type of work will give more scope for further research.

Reviewer #3: Editor:

Please ensure the results are accurately reported, any overstated conclusions are rewritten and the limitations of the work fully explained.

Comments to the Authors

Dear Editor

PLOS ONE

Review report on manuscript #: PONE-D-24-39990R1

Title: “Determination of radiocarbon in environmental objects”

The manuscript is significant, this study focuses on Determination of radiocarbon in environmental objects. The research focused on the radiocarbon distribution in natural components at the 'Degelen' test location within the Semipalatinsk Test Site, known for underground nuclear tests. Samples of soil, plants, and water were collected from areas with water inflows. Using a highly sensitive alpha-beta radiometer SL-300 and the Pyrolyser-6 Trio system for sample preparation, the study found that radiocarbon was nonuniformly distributed in the environmental objects.

The introduction presents the specific work's aim in a well-organized manner. The experimental results were tabulated and illustrated in a comprehensive manner. Both abstracts and conclusions are informative and suitably formed.

After making the revisions suggested by the previous reviewers, the research is now suitable for publication after the corrections below.

Comment 1: Language and Style: The writing is mostly clear, but there are areas where grammatical improvements could enhance readability. A thorough proofreading is recommended.

Comment 2: The conclusion is poor. It should be rewritten according to the significance of the present outcomes in the context. The authors should mention how to achieve the research question mentioned at the end of the introduction. Conclusions must be deeper.

7. PLOS authors have the option to publish the peer review history of their article (what does this mean? ). If published, this will include your full peer review and any attached files.

**Do you want your identity to be public for this peer review?** For information about this choice, including consent withdrawal, please see our Privacy Policy .

Reviewer #1: No

Reviewer #3: No

---

## [Editor Report · Acceptance letter]

PONE-D-24-39990R1

PLOS ONE

Dear Dr. Aidarkhanova,

I'm pleased to inform you that your manuscript has been deemed suitable for publication in PLOS ONE. Congratulations! Your manuscript is now being handed over to our production team.

Kind regards,

on behalf of

Dr. Mohamed Y.M. Hanfi

Academic Editor

PLOS ONE